# The Suspended Sediment Flux in Winter in the South of Chengshantou, between the North and South Yellow Sea

**DOI:** 10.3390/s23187771

**Published:** 2023-09-09

**Authors:** Bowen Li, Xuejun Xiong, Baichuan Duan, Daolong Wang, Long Yu

**Affiliations:** 1College of Oceanography and Space Information, China University of Petroleum, Qingdao 266580, China; 2First Institute of Oceanography, Qingdao 266061, China; xiongxj@fio.org.cn (X.X.); duanbch@fio.org.cn (B.D.); wdl@fio.org.cn (D.W.); ylong@fio.org.cn (L.Y.)

**Keywords:** tides, current, turbidity, suspended sediment flux, Yellow Sea

## Abstract

Due to the regional differences between the North and South Yellow Sea, and under the influence of winter winds, the relative changes in the coastal current and the Yellow Sea warm current will lead to the instability of the front, which will lead to the cross-front transport of sediment. Therefore, the study of sediment exchange between the North and South Yellow Sea has become an indispensable part of the study of the Yellow Sea environment. In this study, the current field and sediment concentration in the southern part of Chengshantou, a representative area of the Yellow Sea, were observed in winter in order to analyze the sediment exchange process between the North Yellow Sea and the South Yellow Sea in winter. The observation results show that in the southern sea area of Chengshantou, in winter, the current velocity does not change with the water depth when it exceeds 15 m, and the tides are regular semi-diurnal tides. When the water depth is less than 15 m, the current direction changes clockwise with the increase in the water depth. The turbidity increases rapidly when the wind direction is offshore and the bottom residual current is onshore, which may cause the sediment transported offshore under the action of wind and ocean current to settle under the obstruction of the Yellow Sea warm current, resulting in the rise of bottom turbidity. This also indicates that the change in residual current direction at different water depths may also lead to an increase in suspended sediment concentration. Based on this, in the estuarine area, the relative change in the current direction between the wind current and the coastal current may also be the cause of the change in the maximum turbidity zone.

## 1. Introduction

The Yellow Sea coast is characterized by a dense concentration of population, industry, and agriculture. There are rich sources of nutrients and extremely rich biological reserves. However, the marine environment of the Yellow Sea is greatly affected by human activities and land-based processes, with complex biogeochemical processes taking place [1]. This is because a large amount of suspended sediment with nutrients is transported by atmospheric sedimentation, river runoff, and the circulation of biogenic elements within the water body to the Yellow Sea [2,3]. Moreover, many rivers are directly or indirectly injected into the sea, so the biogeochemical process is complex [4].

The relationship between source and sink is key to the study of marginal sea interaction, sea level change, and sedimentation. One of the characteristics of marginal seas is the dual terrigenous sedimentary system [5]. The Yellow Sea is backed by the Chinese mainland, and the suspended sediment is mainly composed of terrigenous detritus, mainly from the sediment transport of the new and old Yellow River, Yangtze River, Jinjiang River, Rongshan River, and other rivers [6,7,8,9]. The suspended sediment in the central part of the South Yellow Sea is the sediment transported by the modern Yellow River and the northern Jiangsu shoal (the Laohuang estuary), while the suspended sediment in the South Yellow Sea is mainly the sediment transported by the Jinjiang River and Rongshan River on the Korean Peninsula [6,10]. The analysis of water turbidity in the Bohai Strait and the North Yellow Sea has found that the concentration of suspended solids in the Yellow Sea is relatively high, showing the characteristics of a high nearshore and low farshore, and a high lower layer and low upper layer. The distribution of wind, warm currents in the Yellow Sea, coastal currents, cold water mass, and terrain are important factors affecting the distribution of suspended solids and have been studied in some detail [11,12,13,14,15]. Wave and tide are the main factors that determine the horizontal distribution of suspended bodies in the Yellow Sea [16,17,18,19].

The concentration distribution of suspended sediments in the Yellow Sea has obvious seasonal and regional differences, and the macro transport pattern of suspended sediments in the Yellow Sea has the seasonal characteristics of “storage in summer and transportation in winter” [20]. The concentration of suspended sediment in summer is far lower than that in winter, and the concentration in the middle of the Yellow Sea is far lower than that in the coastal area. Because the strong wind in winter strengthens the warm current in the Yellow Sea and the coastal current in northern Shandong, and greatly improves their ability to carry and efficiency in carrying the suspended sediment, the transport flux of the suspended sediment in summer is far lower than that in winter [21,22]. In addition, the coastal current and the warm current of the Yellow Sea in winter form a strong current shear front at the eastern side of the Shandong Peninsula, which hinders the transport of the suspended sediment to the east [15,23], causing the suspended sediment to become concentrated along the coast and difficult to transport across the front [24]. The suspended sediment in the northern Jiangsu shoal area is also transported to the deep-water area in the central Yellow Sea in autumn and winter [25,26].

The resuspension of seafloor surface sediments caused by waves and currents in winter may be the cause of the suspended sediment layer in the Yellow Sea, which is located below the pycnocline [27,28]. The formation of the layer is also affected by the cold water mass and the thermocline in the central Yellow Sea [29]. Previous studies have shown that there are also significant differences in the composition of suspended matter in the Yellow Sea water in winter and summer [16,22,30]. In summer, there is a locally high concentration of suspended solids near the surface in the deep-water area of the central Yellow Sea, which is mainly caused by the enrichment of plankton [29], and this component consists of more than mineral particles [22]. In winter, resuspension is significant in shallow water areas, and the main components of high-concentration suspended solids are fine-grained sediments under resuspension [14].

Moreover, there are obvious regional differences in the sedimentary dynamics and biogenic factors between the North and South Yellow Sea. The formation of the muddy sedimentary system in the central part of the South Yellow Sea is closely related to the weak current and upwelling, and is a cyclonic cold vortex sedimentary system developed under the low-energy sedimentary environment [31]. Therefore, it can be considered that the formation of and change in the circulation system of the modern Yellow Sea and the change in the basin environment (such as the East Asian monsoon, human activities, and the diversion of the Yellow River) are the two main controlling factors affecting the sedimentary environment and material sources in the central part of the South Yellow Sea [32]. In the southern part of the Yellow Sea, dissolved oxygen is mainly controlled by the warm water of the Yellow Sea and the front water of the Taiwan Warm Current [33,34]; the oxygen is in a state of net consumption, and the amount of CO_2_ absorbed is greater than the amount released, which leads to the gradual acidification of the Yellow Sea [35]. In summer, the dissolved oxygen level is high in the northwest and southeast and low in the middle [36]. Under the effect of the transport of fresh water from the Yangtze River, the nutrient content (especially the nitrate content) in the waters in the southwest of the South Yellow Sea and the northeast of the Yangtze River estuary presents high values [37,38] and also shows an upward trend [39].

Due to the regional differences between the North and South Yellow Sea, and under the influence of winter winds, the relative changes in the coastal current and the Yellow Sea warm current will lead to the instability of the front, which will lead to the cross-front transport of sediment [40]. Therefore, the study of sediment exchange in the North and South Yellow Sea has become an indispensable part of environmental research in the Yellow Sea. At present, two methods of in situ observation and numerical simulation are used to study the water and sediment exchange process in two different areas. The former is mostly used for the observation of the sediment transport into the sea, and it is necessary to observe the water turbidity and velocity of the current at the same time [41]. The latter uses a model calculation, and the calculation results need to be compared with the observation data to determine whether the adopted model is suitable for the simulated area [42]. Due to the lack of rich sediment exchange data between the North and South Yellow Sea, in this paper, the sediment exchange in the North and South Yellow Sea is studied by in situ observation.

Chengshantou is located to the east of the Shandong Peninsula, surrounded by the sea on three sides, and the water depth outside the corner changes sharply. The connection between Chengshantou and Changshangot in the Korean Peninsula divides the Yellow Sea into two parts: the south and the north. Therefore, in this paper, the current field and sediment concentration in the southern part of Chengshantou, a representative area of the Yellow Sea, were observed in winter in order to analyze the sediment exchange process between the North Yellow Sea and the South Yellow Sea in winter. This work not only enriches the data concerning the material exchange process in the North and South Yellow Sea but also contributes to the study of the overall sediment cycle in the Yellow Sea. It is also helpful to analyze the source of nutrients in the North and South Yellow Sea and provide a basis for the management of fishery production.

## 2. Methods

The study area is located in the northern part of Chengshantou (122.54° E, 36.55° N) (Figure 1). The water depth of the sea area 1 km offshore can reach 50 m. Its special geographical location and topographic structure make the tidal current velocity of the sea area very high, and the measured velocity is as high as 2 m/s [43]. This sea area belongs to the irregular semidiurnal tidal zone, which is very close to the M2 and S2 non-tidal points in the Yellow Sea, with a small tidal range, and an annual average tidal range of about 0.8 m. However, the tidal current in this sea area is the regular semidiurnal tidal zone, and the current direction is from northeast to southwest.

In this study, an ADCP mounted on a buoy floating on the water surface was used to observe current profiles, and a turbidimeter mounted on a tripod attached to the buoy to observe the resuspended sediment concentration near the bottom bed (the distance above the seabed is about 0.6 m) in the southern part of the Chengshantou area from 8 November 2020 to 8 February 2021 (Figure 1). The carrying equipment is shown in Table 1.

## 3. Results

During the period from 8 November 2020 to 8 February 2021, the current velocity in the southern part of Chengshantou fluctuated mainly in the range of 0–0.8 m/s, and the water depth was approximately 30 m (Figure 2). During the observation period, the current direction profile was mainly divided into two layers, and the velocity profile into three layers. The large change in the velocity direction of the surface current is mainly affected by wind and wave currents in the upper layer, while the small change in the velocity direction of the bottom current is mainly due to the tidal current in the bottom layer (Figure 3). The middle layer of the velocity profile may be the transition layer between the two layers because the current velocity is low (Figure 2). In winter, the current velocity at different depths of the lower layer was basically consistent with the current direction in a vertical direction (Figure 2). Therefore, the vertical velocity profile at one time point in each month was selected for analysis (Figure 3). The current velocity profile at the three time points (one before the increase in turbidity, one during the increase in turbidity, and one after the turbidity reaches the limit (Figure 4)) in Figure 3 shows that the current velocity remained basically unchanged on November 17 when the water depth was less than 15 m, and fluctuated between 0.15 ± 0.05 m/s on December 17 and 0.20 ± 0.05 m/s on January 9. When the water depth was less than 15 m, the velocity increased rapidly to a maximum of more than 2 m/s (Figure 3a,c,e). Similarly, when the water depth exceeded 15 m, the current direction basically did not change with the water depth. When the water depth was less than 15 m, the current direction changed clockwise with the increase in the water depth (Figure 3b,d,f).

When the distance from the bottom bed is more than 15 m, the current velocity and direction hardly affect the seabed. Therefore, the deep mean current velocity with water depth exceeding 15 m was used to analyze the change in turbidity near the seabed in the southern part of the Chengshantou Sea area (Figure 4). The changes in the tidal current velocity, current direction, and turbidity in winter are shown in Figure 5. Before Nov. 28th, the turbidity in this area was less than 50 NTU, rose to more than 120 NTU after Dec. 1st, and dropped to approximately 90 NTU after two months (Figure 4). The velocity and direction of the current also changed during the turbidity increase. Therefore, Figure 5 was drawn according to the changes in the tidal current direction and the velocity before the rise. Figure 5 shows a regular semi-diurnal tide in the southern part of the Chengshantou sea area; its rapid current velocity during the rise and fall of the tide was 0.30–0.50 m/s, and its slack water velocity was less than 0.10 m/s.

## 4. Discussion

### 4.1. Suspended Sediment Fluxes

The change in the sediment flux prior to the turbidity increase is shown in Figure 6. The variation trend of the sediment flux is similar to that of the current velocity, showing periodic changes, and the maximum sediment flux reached 25 NTU × m/s. After the increase, the turbidity and suspended sediment fluxes increased more than three times (Figure 7). However, in comparing the velocity and turbidity changes, no obvious correlation was found between turbidity and velocity near the bottom bed (Figure 7).

### 4.2. Resuspended Sediment Transport

Since the rapid increase in turbidity may not be caused by local sediment resuspension, the role of sediment transport needs to be considered. Therefore, we analyzed the changes in the residual current during the observation period and found that the local residual current is mainly divided into upper and lower layers, and the partial residual current velocity is divided into three layers, which is similar to the stratification of the overall velocity (Figure 8). When the turbidity increases rapidly, the bottom residual flow velocity increases, the flow direction is from 270° to 360°, and the residual current direction is toward the shore (Figure 1 and Figure 8). It may be that the residual current obstructs the seagoing diffusion of terrigenous sediments, leading to the increase in suspended sediment concentrations in this area.

### 4.3. Current Stratification

The tidal current and residual current in the observation area are mainly divided into two layers. The change in the upper current direction is similar to the change in the wind direction (go direction), and the upper residual current may be mainly affected by the wind current (Figure 9). Compared with the wind direction before and after the rapid increase in turbidity and the time of decline, the turbidity is larger when there is a stronger westerly wind (when the wind blows eastwards) (Figure 10). When the turbidity increased rapidly, the wind direction was mainly offshore and the bottom residual current was onshore, which may have caused the sediment transported offshore under the action of the wind and ocean currents to settle under the obstruction of the bottom residual current, resulting in the rise in the bottom turbidity.

In order to analyze whether the bottom current is affected by the Yellow Sea warm current, the changes in the air temperature, surface water temperature, and bottom water temperature during the observation period were analyzed. It was found that the changes in the surface water temperature and bottom water temperature were basically the same, and there was no obvious decrease when the temperature dropped greatly (Figure 11a). In order to exclude the influence of diurnal temperature difference, the average daily air temperature and surface water temperature were compared (Figure 11b). It was found that while the daily average temperature dropped significantly, the water temperature did not; therefore, it can be inferred that the current in this area may be affected by the Yellow Sea warm current (Figure 11b).

## 5. Conclusions

The results show that the current velocity does not change when the water depth exceeds 15 m, and the range of variation is less than 5 cm/s. The tide is a regular semi-diurnal tide, with a rapid current velocity during the rise and fall of the tide of 0.30–0.50 m/s and a slack water velocity of less than 0.10 m/s. When the water depth is less than 15 m, the current direction changes clockwise with the increase in the water depth, and the velocity increases to 2 m/s. The variation trend of the sediment flux is similar to that of the current velocity, showing periodic changes; the maximum sediment flux can reach 25 NTU × m/s before the turbidity increases rapidly. 

In winter, the surface residual current is mainly affected by the wind current, while the bottom current is affected by the Yellow Sea warm current. The turbidity increases rapidly when the wind direction is offshore and the bottom residual current is onshore, which may cause the sediment transported offshore under the action of the wind and ocean currents to settle under the obstruction of the Yellow Sea warm current, resulting in an increase in the bottom turbidity.

The results also indicate that in addition to the sediment resuspension caused by current velocity and tidal current velocity, the change in residual current direction at different water depths may also lead to an increase in suspended sediment concentration. Therefore, it is necessary to use current profile observation instead of single-point current observation in some sea areas to clarify the mechanism of suspended sediments and the ecological environment change. Especially in the estuarine area, the relative change in the current direction between the wind current and the coastal current may also be the cause of the change in the maximum turbidity zone.

## Figures and Tables

**Figure 1 sensors-23-07771-f001:**
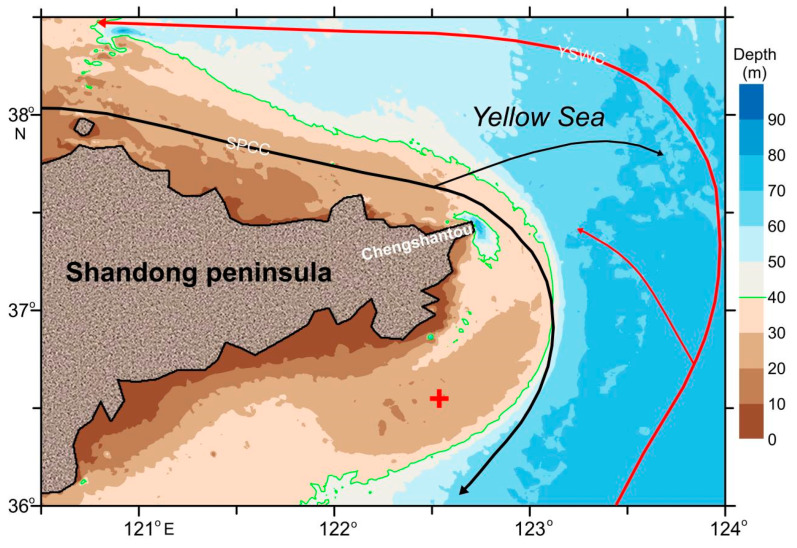
Location of the observation point (122.537° E, 36.549° N). The red cross is the observation point position. The black arrows (SPCC) is the Shandong peninsula coastal current and the red arrows (YSWC) is the Yellow Sea warm current [44].

**Figure 2 sensors-23-07771-f002:**
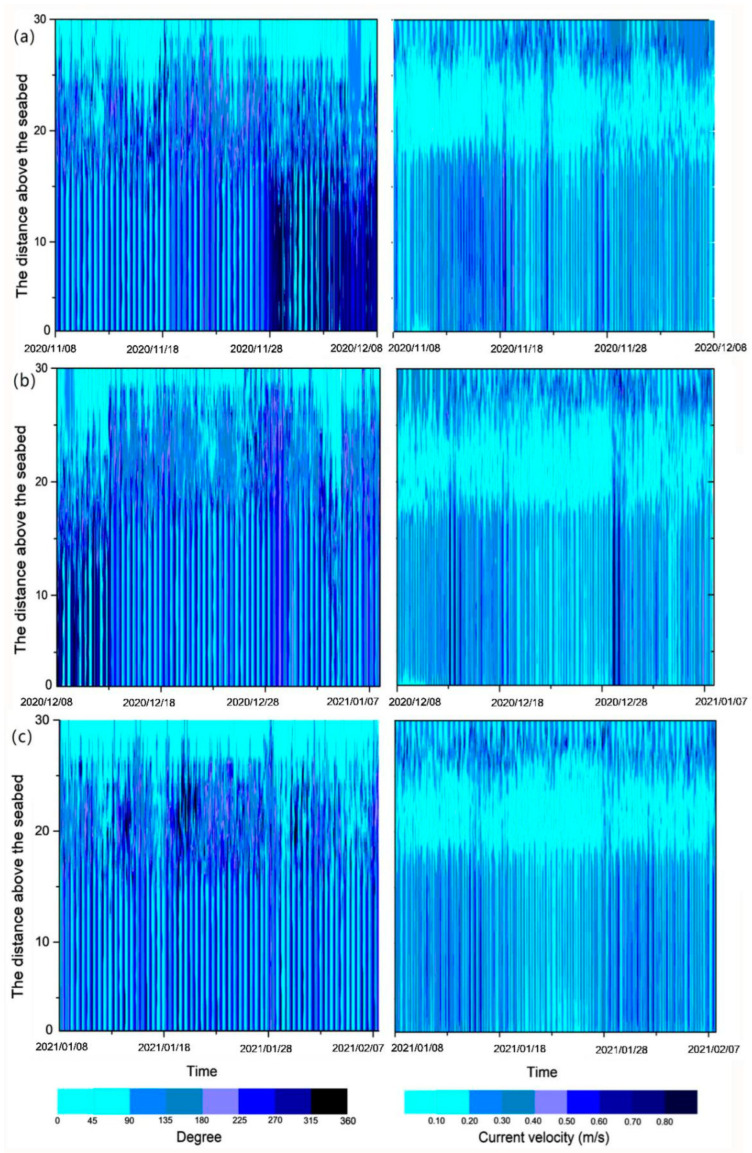
Current degree and current velocity profile from (**a**) 8 November to 8 December 2020, (**b**) 8 December 2020 to 7 January 2021, and (**c**) 8 January to 7 February 2020, in the south of Chengshantou.

**Figure 3 sensors-23-07771-f003:**
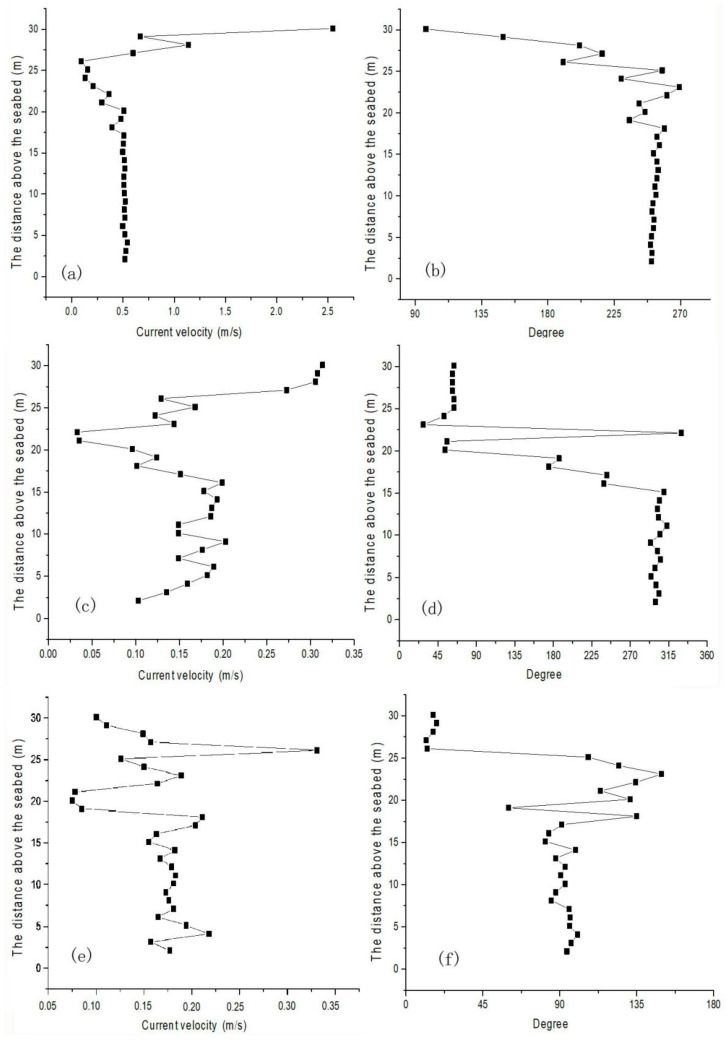
Vertical variation in current velocity and degree in November, December, and January. (**a**) Current velocity at 17:00 on 17 November 2020. (**b**) Current degree at 17:00 on 17 November 2020. (**c**) Current velocity at 10:00 on 17 December 2020. (**d**) Current degree at 10:00 on 17 December 2020. (**e**) Current velocity at 2:00 on 9 January 2021. (**f**) Current degree at 2:00 on 9 January 2021.

**Figure 4 sensors-23-07771-f004:**
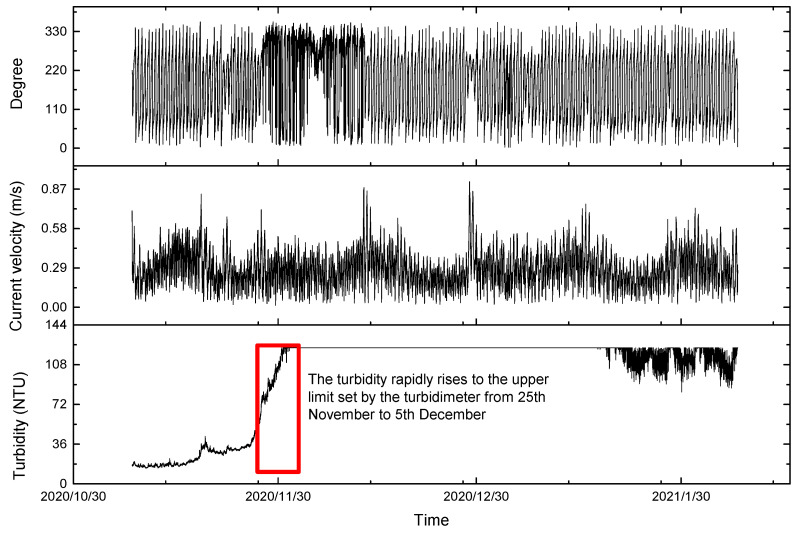
Turbidity (**bottom**), average current velocity (**middle**), and degree (**top**) from 8 November 2020 to 7 February 2021 in the south of Chengshantou.

**Figure 5 sensors-23-07771-f005:**
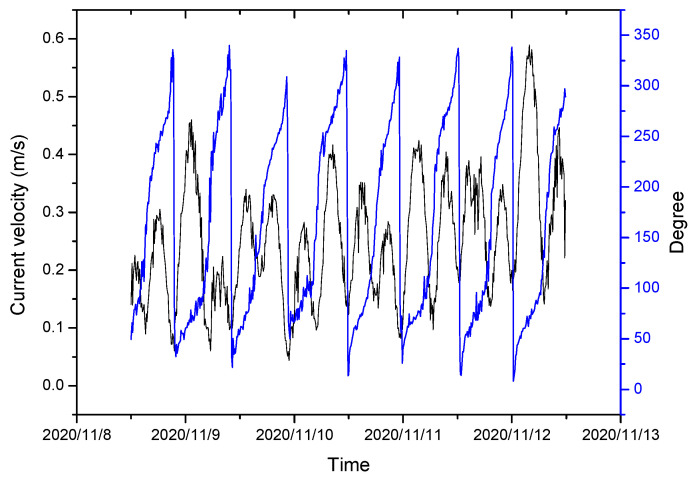
Current velocity varied with the current direction from 8 November 2020 to 12 November 2020.

**Figure 6 sensors-23-07771-f006:**
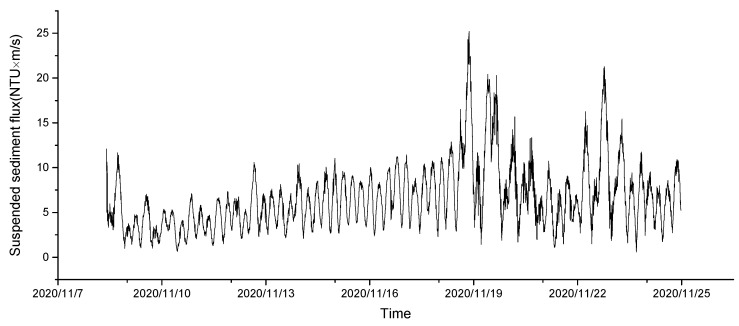
Suspended sediment flux before the turbidity increase.

**Figure 7 sensors-23-07771-f007:**
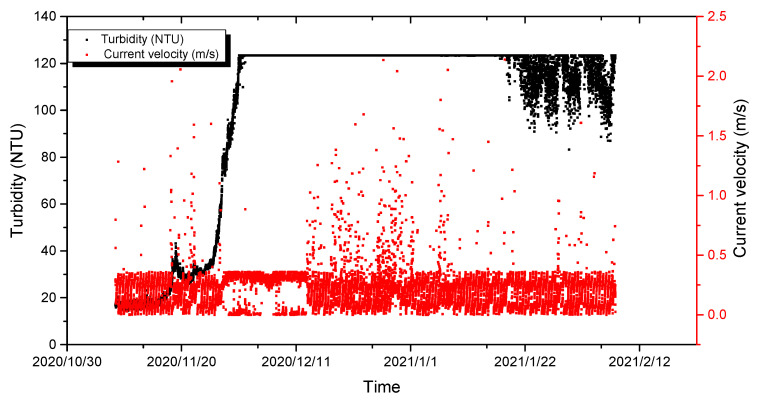
Turbidity varied with the current velocity near the seabed from 8 November 2020 to 7 February 2021.

**Figure 8 sensors-23-07771-f008:**
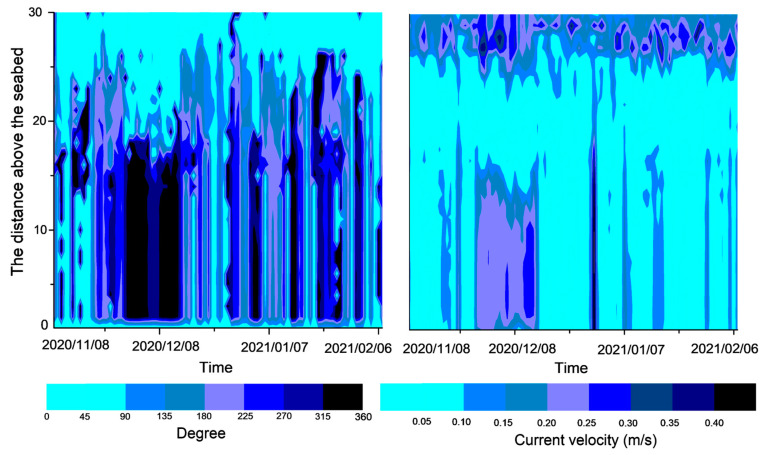
Residual tidal current degree and current velocity profile from 8 November 2020 to 7 February 2021 in the south of Chengshantou.

**Figure 9 sensors-23-07771-f009:**
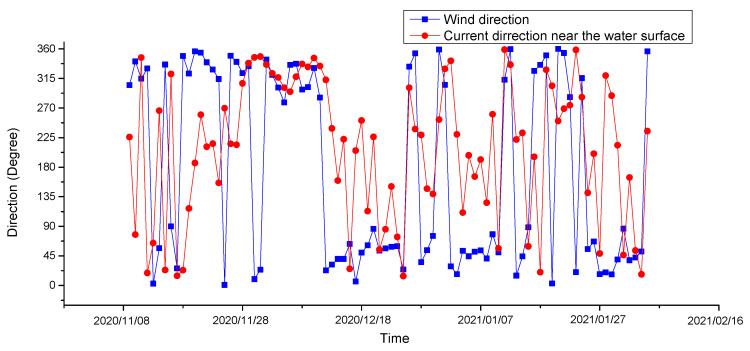
The direction of the wind (go direction) and current near the water surface from 8 November 2020 to 7 February 2021 in the south of Chengshantou.

**Figure 10 sensors-23-07771-f010:**
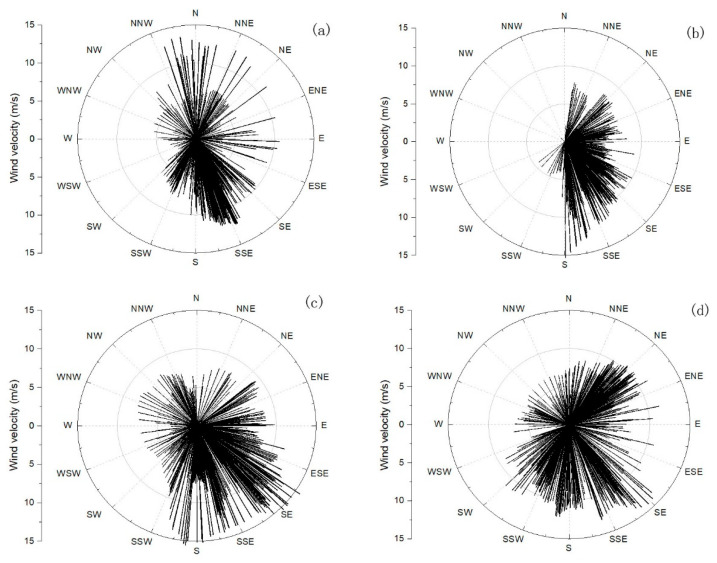
Wind velocity and direction (go direction) from 8 November 2020 to 8 February 2021 in the south of Chengshantou. (**a**) Wind velocity and direction from 8 to 25 November 2020. (**b**) Wind velocity and direction from 26 November to 20 December 2020. (**c**) Wind velocity and direction from 21 December 2020 to 19 January 2021. (**d**) Wind velocity and direction from 20 January to 8 February 2021.

**Figure 11 sensors-23-07771-f011:**
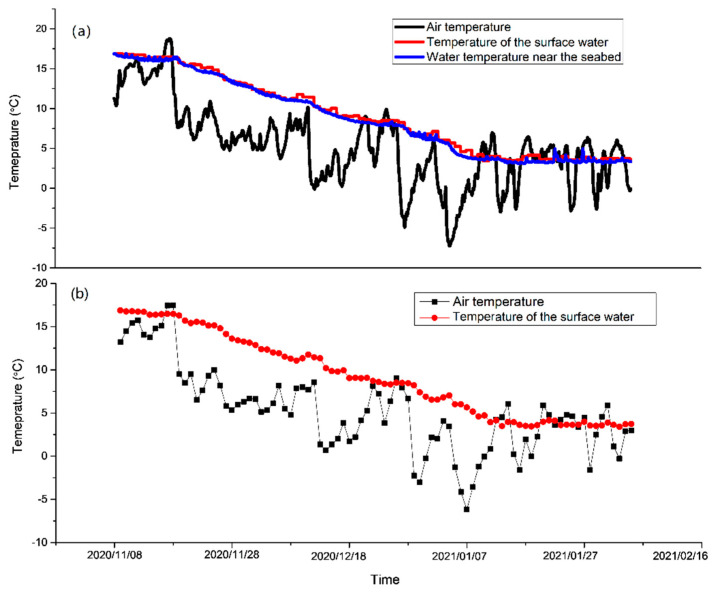
Temperature change in air, surface water, and water (**a**) and temperature of the surface water changed with averaged air temperature near the seabed from 8 November 2020 to 8 February 2021 in the south of Chengshantou.

**Table 1 sensors-23-07771-t001:** Summary table of equipment parameters.

Equipment	Main Function	Main Index
Temperature and salt depth turbidimeter	Observes parameters such as temperature, salinity, pressure, turbidity, dissolved oxygen, and oxygen redox potential (ORP)	The accuracy of the temperature and salt sensor is 0.002 degrees; the salinity accuracy is 0.003 ms/cm. The accuracy of the pressure sensor is 0.05% of the water depth. The accuracy of the optical dissolved oxygen sensor is 5%. The accuracy of the ORP sensor is 0.01 V. The accuracy of the turbidity sensing is 2%.
600 K HZ ADCP	Velocity profile (upward)	The blind zone is 2.11 m. The distance is 2 m per layer and 17 frequencies are emitted every 600 s.

## Data Availability

The data presented in this study are available on request from the corresponding author. The data are not publicly available due to the data takes up too much space.

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
