# Peer review of "The Suspended Sediment Flux in Winter in the South of Chengshantou, between the North and South Yellow Sea"

_sensors, 2023, doi:10.3390/s23187771_

Round 1
Reviewer 1 Report
The current field and sediment concentration in the southern part of Chengshantou were observed in winter in order to analyze the sediment exchange process between the North Yellow Sea and the South Yellow Sea in winter. Generally, it is a well organized and can provide useful experience for other similar studies. How, some key problems are not very clear and need major revisions. The detail comments are list below:1. Introduction: The authors should describe what is known about the study method and observation technology for flow and the suspended sediment flux and what is lacking in existing study. The research gap is not clearly defined. Some references need to be included in this study.
For example, Hu Z, Guo K, Yang Y and Zhang M (2023) Field survey and analysis of water flux and salinity gradients considering the effects of sea ice coverage and rubber dam: a case study of the Liao River Estuary, China. Front. Mar. Sci. 10:1154150. doi: 10.3389/fmars.2023.1154150
2. Introduction: A large number of the study of biogenic factors in the North Yellow Sea were introduced by the authors, I don't think it has much relevance to this manuscript.
3. The English should be checked and improved entirely.
Line147: and one after the turbidity reaches the limit (Fig. 4)) in Fig. 3 shows that the current velocity remained basically unchanged
line149:When the water depth less than 15 m, the
line 152: When the water depth less than 15 m,
The accuracy of temperature and salt sensor is 0.002 degrees; “salt sensor”
……
4. Figure4, vellocity should be velocity.
5. I don’t understand this sentence “Since the water depth is more than 15 m, the current velocity and direction could affect the seabed”.
6. Please explain the reason of the vertical variation of current velocity and degree in Fig.4.
7. The points in Figure 5 are difficult to identify, and I recommend to carry out optimization.
8. This paper uses the equipment mounted on the float to observe resuspended sediment and changes of current in the south part of Chengshantou area from Nov. 8, 2020 to Feb. 8, 2021 (Fig.1b). The tripod and its carrying equipment are shown in the tab. 1. This is acceptable because the readers do not know the details of the measurements, the parameters of the instrument, the sampling details and the data, as well as the data processing. For example, how were the data in the top and bottom layers (blind zone) treated?
9. It is suggested to propose some more general conclusions or comments for other researchers or administrators to make decisions when facing the similar circumstances, rather than for Chengshantou waters. Thus, the conclusions and abstract can be improved.
10. In Figure 9, does the author consider wind speed in addition to wind direction?
11. Delete (B) in Figure 11.
12. How is the suspended sediment flux calculated in this study? Why was the suspended sediment concentration not used for the flux calculation? Please explain it.
Author Response
Response to Reviewer 1 Comments
The current field and sediment concentration in the southern part of Chengshantou were observed in winter in order to analyze the sediment exchange process between the North Yellow Sea and the South Yellow Sea in winter. Generally, it is a well organized and can provide useful experience for other similar studies. How, some key problems are not very clear and need major revisions. The detail comments are list below:
Response: Thank you for your revision. I have response to your question point by point.
Point 1: Introduction: The authors should describe what is known about the study method and observation technology for flow and the suspended sediment flux and what is lacking in existing study. The research gap is not clearly defined. Some references need to be included in this study.
For example, Hu Z, Guo K, Yang Y and Zhang M (2023) Field survey and analysis of water flux and salinity gradients considering the effects of sea ice coverage and rubber dam: a case study of the Liao River Estuary, China. Front. Mar. Sci. 10:1154150. doi: 10.3389/fmars.2023.1154150
Response1: Thank you for your suggestion. I have added the study method and observation technology for current and the suspended sediment flux and references in lines 94-107. You could consider the following reference.
Reference:
[41] Hu Z, Guo K, Yang Y, et al., 2023. Field survey and analysis of water flux and salinity gradients considering the effects of sea ice coverage and rubber dam: a case study of the Liao River Estuary, China. Front. Mar. Sci., 10, 1154150.
[42] Zhang M, Xu Y, Qiao H, 2018. Numerical Study of Hydrodynamic and Solute Transport with Discontinuous Flows in Coastal Water. Environ Model Assess, 23, 353-367.
Point 2: Introduction: A large number of the study of biogenic factors in the North Yellow Sea were introduced by the authors, I don't think it has much relevance to this manuscript.
Response 2: Thank you for your suggestion. I have delete the paragraph about the biogenic factors in the North Yellow Sea. The delete paragraph is as follows:
Compared with the study of the South Yellow Sea, the study of biogenic factors in the North Yellow Sea is relatively small. The main factors controlling nutrients in the North Yellow Sea are the cold water mass and biological activities, and the impact of land runoff is not significant [40]. In contrast, the South Yellow Sea cold water mass has the characteristics of relatively high temperature, high salinity, low oxygen, high SiO3-Si and high DIN, while the North Yellow Sea cold water mass has the characteristics of low salinity, high oxygen, low pH, low suspended particulate matter, low SiO3-Si and low DIN. The bottom water of the two water masses has high values of PO4-P, SiO3-Si and DIN, and the content of PO4-P is the highest in the whole Yellow Sea area. There is a good positive correlation between the three types of nutrients in the South Yellow Sea cold Water Mass and a significant negative correlation with DO, while the correlation between the North Yellow Sea Cold Water Mass is not significant [41].
Point 3: The English should be checked and improved entirely.
Line147: and one after the turbidity reaches the limit (Fig. 4)) in Fig. 3 shows that the current velocity remained basically unchanged
line149:When the water depth less than 15 m, the
line 152: When the water depth less than 15 m,
The accuracy of temperature and salt sensor is 0.002 degrees; “salt sensor”
……
Response 3: Thank you for your suggestion. I have revised the language. The related certificate is as follows. The specific changes can be seen in the english-edited-70239.
Point 4: Figure4, vellocity should be velocity..
Response 4: Thank you for your suggestion. I have revised vellocity to velocity in Figure 4.
Point 5: I don’t understand this sentence “Since the water depth is more than 15 m, the current velocity and direction could affect the seabed”.
Response 5: Thank you for your suggestion. It is mainly expressed that waves and currents near the bottom bed can affect the erosion and re-suspension of bottom bed sediments. It has been changed to when the distance from the bottom bed is more than 15m, the current velocity and direction are difficult to affect the seabed. The related sentences are revised in Lines 168-169.
Point 6: Please explain the reason of the vertical variation of current velocity and degree in Fig.4.
Response 6: Thank you for your suggestion. The vertical change of velocity and direction may be due to the large change of velocity direction of surface current, which is mainly affected by wind and wave currents in the upper layer, while the small change of velocity direction of bottom current is mainly tidal current in the bottom layer, and the middle layer is the transition zone of both. This sentence is added in Lines 144-146.
Point 7: The points in Figure 5 are difficult to identify, and I recommend to carry out optimization.
Response 7: Thank you for your suggestion. Figure 5 shows a line chart with no points. If you are talking about Figure 7, Figure 7 is mainly to illustrate the correlation between current velocity and turbidity. Obviously, turbidity (black point) and current velocity (red point) in Figure 7 cannot be fitted, and there is no obvious correlation between them. The related sentences are located in Lines 192-193. And I've narrowed down the points in Figure 7 to help you see them more clearly.
Point 8: This paper uses the equipment mounted on the float to observe resuspended sediment and changes of current in the south part of Chengshantou area from Nov. 8, 2020 to Feb. 8, 2021 (Fig.1b). The tripod and its carrying equipment are shown in the tab. 1. This is acceptable because the readers do not know the details of the measurements, the parameters of the instrument, the sampling details and the data, as well as the data processing. For example, how were the data in the top and bottom layers (blind zone) treated?
Response 8: Thank you for your suggestion. This paper uses the ADCP mounted on a buoy floating on the water surface to observe profiles of current and turbidimeter mounted on a tripod attached to the buoy to observe the resuspended sediment concentration near the bottom bed (the diatance above the seabed is about 0.6 m). These sentences are added in Lines 129-132.
Only ADCP has a blind area. The blind area of ADCP is near the water surface. The data displayed is the current direction data within 30m from the bottom bed, and the data not observed in the blind area is not displayed.
Point 9: It is suggested to propose some more general conclusions or comments for other researchers or administrators to make decisions when facing the similar circumstances, rather than for Chengshantou waters. Thus, the conclusions and abstract can be improved.
Response 9: Thank you for your suggestion. I have added new conclusions or comments in abstract (Lines 18-21) and conclusions (258-265).
Point 10: In Figure 9, does the author consider wind speed in addition to wind direction?
Response 10: Thank you for your question. In this paper we have taken into account wind direction and speed. For example, we analyze that the west wind is stronger and the wind direction is mainly offshore in Lines 217-218, based on wind speed in different directions (Fig. 9).
Point 11: Delete (B) in Figure 11.
Response 11: Thank you for your suggestion. Figure 11a is the measured temperature per hour. And Figure 11b is the average temperature per day, in order to eliminate the influence of diurnal temperature difference. The related explanations are showen in Lines 230-232.
Point 12: How is the suspended sediment flux calculated in this study? Why was the suspended sediment concentration not used for the flux calculation? Please explain it.
Response 12: Thank you for your question. The suspended sediment flux is calculated by multiplying turbidity by current velocity.
The suspended sediment concentration is not calculated because the observed suspended sediment concentration is generally obtained by correcting the observed turbidity with the water sample collected in situ. In this observation, no real-time water collection is performed, so turbidity rather than suspended sediment concentration is used to calculate the suspended sediment flux.

Reviewer 2 Report
This is an interesting case report, negatively impacted by the fact the English is (very) poor.
The English is poor to incomprehensible. Very significant improvements are needed.
Author Response
Thank you for your suggestion. I have added new conclusions or comments in abstract (Lines 18-21) and conclusions (258-265).
I have revised the language. The related certificate is as follows. The specific changes can be seen in the english-edited-70239.

Round 2
Reviewer 1 Report
The reviewer has no additional comments here.